# Alterations of Content and Composition of Individual Sulfolipids, and Change of Fatty Acids Profile of Galactolipids in Lettuce Plants (*Lactuca sativa* L.) Grown under Sulfur Nutrition

**DOI:** 10.3390/plants11101342

**Published:** 2022-05-18

**Authors:** Tania T. Körber, Noah Frantz, Tobias Sitz, Muna A. Abdalla, Karl H. Mühling, Sascha Rohn

**Affiliations:** 1Institute of Food Chemistry, Hamburg School of Food Science, University of Hamburg, Grindelallee 117, 20146 Hamburg, Germany; tania.koerber@chemie.uni-hamburg.de (T.T.K.); noah.frantz@studium.uni-hamburg.de (N.F.); tobias.sitz@chemie.uni-hamburg.de (T.S.); 2Institute of Plant Nutrition and Soil Science, Kiel University, Hermann-Rodewald-Str. 2, 24118 Kiel, Germany; mabdalla@plantnutrition.uni-kiel.de (M.A.A.); khmuehling@plantnutrition.uni-kiel.de (K.H.M.); 3Institute of Food Technology and Food Chemistry, Technische Universität Berlin, Gustav-Meyer-Allee 25, 13355 Berlin, Germany

**Keywords:** lettuce, sulfur, abiotic stress, hydroponics, sulfolipids, fatty acids, chlorophyll a, chlorophyll b, carotenoid

## Abstract

Alterations of chloroplast membrane lipids might serve as indicators of eco-physiologically induced and plant nutrition-induced changes during plant growth. The change in the degree of fatty acid saturation in the membranes is in particular a strategy of plants to adapt to abiotic stress conditions. Green multi-leaf lettuce plants (*Lactuca sativa* L.) were subjected to three different sulfur (S) levels. Sulfoquinovosyl diacylglycerol derivatives (SQDG) might be affected by S nutrition. Therefore, the present study was conducted to investigate the impact of S fertilization on the content and composition of individual SQDG. In addition to a change in the SQDG composition, a general change in the total lipid composition of the chloroplast membrane was observed. A significant increase in total SQDG content and doubling of the galactolipid content and significant alterations of individual SQDG were observed at elevated levels of S fertilization. High levels of S supply demonstrated a clear trend of increasing total chloroplast lipid content and concentrations of linolenic acid, in addition to a further decline in palmitic acid. The study opens perspectives on S supply and its crucial role in the build-up of photosynthetic apparatus. Moreover, it emphasizes the role of S-containing compounds, including sulfolipids, in modulating physiological adjustment mechanisms to improve tolerance ability to various abiotic stresses in plants and, consequently, plant food quality.

## 1. Introduction

Many cultivation conditions such as temperature, solar radiation, nutrient content of the soil, or water availability can have diverse impacts on food plants. Depending on the application of several nutrient elements, effects on the biosynthesis of e.g., proteins, enzymes, vitamins, and pigments can be observed [1]. Consequently, plant food quality is affected, but it also bears the potential to direct the growth and resistance of plants by improved nutrition. Next to an increase in yield, nutrient supply also has an effect on compound composition, which in turn might affect human nutrition and health status. Sulfur (S) is an essential macronutrient for plant growth and metabolism. It is a constituent of amino acids, glutathione (GSH), phytochelatins, chloroplasts, sulfolipids, vitamins, and prosthetic groups (including iron-S centers, thiamine, lipoic acid, coenzyme A, and many more) in addition to S-containing secondary metabolites [2]. S-compounds have demonstrated profound effects on plant disease resistance, in addition to various pharmacological properties and significant health benefits [3]. One prominent example is the sulfoquinovosyl diacylglycerol derivatives (SQDG), which exhibit antiviral effects [4]. Furthermore, sulfonates (e.g., SQDG, taurine, and isethionates) and their final hydrolysis product, hydrogen sulfide (H_2_S), showed various greater physiological effects in the human gut microbiome. For instance, some microorganisms produce H_2_S as a toxic metabolite during biotransformation in the mammalian gut after the consumption of plant food [5,6,7]. Generally, SQDG can be ubiquitously found in photosynthetically active organisms. Their chemical structure is composed of a sulfoquinovose, a diacylglycerol backbone linked with one or two fatty acids, which differ in chain length and the degree of unsaturation. Consequently, and similar to other glycerol-based lipids, many different individual SQDG are possible. Recently, a LC-ESI-MS/MS method was developed for the individual identification and quantification of SQDG [8].

SQDG are important for the structure and function of the thylakoid membrane of plant cell chloroplasts, and they seem to be important for optimal photosynthesis. These membranes have a unique structure and contain three kinds of non-phosphorus glycolipids and phosphatidylglycerols (PG). SQDG and PG can be assigned to anionic lipids and represent around 16% of the thylakoid membrane lipids. Beyond these, most thylakoid membrane lipids are composed of other non-phosphorus glycolipids. The chargeless galactolipids, primarily monogalactosyl diacylglycerol (MGDG) and digalactosyldiacylglycerol (DGDG), account for around 70–85% of the membrane [9,10]. Under conditions of abiotic stress, the photosynthesis apparatus is affected and a decline in plant yield, a decrease in pigment concentration [11], and alterations in the levels of thylakoid membrane lipids occur. As a protection mechanism, there is an alteration in the synthesis and composition of the thylakoid membrane lipids [12,13,14,15,16]. The degree of unsaturation, as well as an exchange of a thylakoid lipid class (e.g., SQDG, PG vs. galactolipids), is outlined in various nutrient starvation situations. For example, the compensation of PG through the enhanced formation of SQDG and galactolipids at a status of phosphor (P) starvation [17] and the degradation of SQDG in the case of S starvation [18] have been shown. A change in the degree of saturation of the fatty acid-bound membrane lipids leads to the better adaptation of membranes to temperature stress [19,20,21]. In this context, salinity, as another stress factor, has also been shown to lead to a reduction of total fatty acids and a decrease in polyunsaturated fatty acids [22]

Being a good model plant, but also one of the most important fresh-cut vegetables in the world, lettuce (*Lactuca sativa* L.) is a decent source of various vitamins and essential ingredients, such as bioactive metabolites for human health [23]. However, nutrient management is a major challenge to cultivation in hydroponic systems [24]. Abdalla et al. [25] already showed that the S application in lettuce cultivated in these systems can increase its yield and dry mass (DM). The use of such systems for the cultivation of lettuce plants has, on the one hand, the benefit of adapting the nutrient supply directly to the plant’s necessities, consequently, providing optimal growth conditions. On the other hand, hydroponic systems are suitable for studying and characterizing transformation reactions resulting from nutrient supply under certain experimental conditions.

The aim of the present study was to characterize the dependence of different levels of the S supply on the formation of individual SQDG. However, SQDG are to some extent neglected metabolites for an evaluation of different sulfur supply levels. Consequently, the focus was on those metabolic influences. As already mentioned, lettuce is an interesting model and used in this case, as there was already a good experience with sulfur supply levels for an appropriate growth [25]. Besides S deprivation, being an important abiotic stress factor (nutrient deficiency) for plants, two other concentrations were chosen to characterize the possible impact of a higher supply of S. Until now, no data about the effects on individual SQDG have been published. When SQDG composition is affected, further glycolipids and their fatty acid composition might be transformed as well. Subsequently, the non-phosphorus glycolipids might serve as markers for different stressors. In the present study, fatty acids of the galactolipids were considered, in order to obtain an overview of the fatty acid composition of the main thylakoid membrane lipids. However, the authors examined the hypothesis that S nutrition might have an influence on the thylakoid membrane especially by affecting SQDG composition.

## 2. Results

### 2.1. Impact of S Application on Different Lipids of the Chloroplast Membrane

To investigate the impact of S nutrition on the potential transformation of thylakoid membrane lipids, individual sulfolipids, galactolipids, and their corresponding esterified fatty acids were analyzed due to their almost exclusive presence in chloroplast membranes. S deprivation (referred to as S0) was used as an abiotic stress factor. However, concentrations of 1 and 1.5 mM K_2_SO_4_ (referred to as S1 and S2) were applied to determine the effects of optimal or excessive S availability, respectively. The chloroplast membranes can be characterized as thylakoid membranes and outer and inner envelopes. Inner envelopes and thylakoids show similar lipid compositions. In contrast, the outer envelope contains relatively less MGDG and around 30% phosphatidylcholine [9]. The thylakoid membrane is essential for photosynthesis and contains the photosystems (PS). Next to the phosphatidylglycerols (PG), glycolipids are important for the structure and function of the chloroplast membranes and for ensuring optimal photosynthetic activity in the thylakoids.

#### 2.1.1. Determination and Composition of SQDG

Primarily, individual SQDG were determined with a targeted LC-ESI-MS/MS method. With the aid of the commercially available standard of the SQDG with a mass-to-charge ratio (*m/z*) in the negative mode of 815 ([M − H]^+^, “SQDG derivative 816”), all sulfolipids were determined semi-quantitatively. The sum of all of the individual SQDG was described as the total content.

S fertilization significantly influenced the composition and content of SQDG in lettuce. An increase in S concentration led to higher total SQDG contents. In comparison to S deprivation (level 0), an adequate S level (S1) induced a 4.8-fold increase. By elevating it to S2, a further significant rise of 26% was detectable (*p* < 0.05). When comparing S deprivation (S0) with S excess (S2), a significant six-fold increase in total SQDG content (2.1 mg/g DM, *p* < 0.001) was observed. The total sulfolipid content in dependence on the S treatment levels is shown in Figure 1. With regard to the sulfolipid composition, 20 different SQDG derivatives were detected. Their *m/z* and the composition of individual SQDG in the lettuce plants grown under varied S levels are presented in Table 1.

The influence of S fertilization was analyzed with regard to the formation of individual SQDG, based on their percentage of occurrence at the different S levels. For this purpose, an enrichment factor for the total SQDG content was determined between S0 as a basis and S2 as a higher concentration (c_1.5mM_/c_0mM_). The mean enrichment factor of all SQDG was 6.0 (2085 µg g^−1^ DM/346 µg g^−1^ DM = 6.0).

To elucidate the individual formation of each sulfolipid, the enrichment factor was calculated for each individual SQDG. Enrichment factors of individual SQDG higher than 6.0 indicate a preferential formation, while a factor below 6.0 suggests a lower enrichment in lettuce fertilized with S (S2 in contrast to S0). For this calculation, the percentual content of the individual SQDG treated with S2 was divided by the content of the respective SQDG at S0 (Table 1). The SQDG affected the most by the S treatments were as follows: *m/z*: 765, 789, 793, 813, and 837. The SQDG 789, 813, and 837 had enrichment factors markedly higher than 6.0, while the SQDG derivatives 765 and 793 had lower factors. A common characteristic of the preferentially formed SQDG was the presence of at least one linked unsaturated fatty acid: in detail, palmitoleic acid (16:1); palmitolinoleic acid (16:2); hexadecatrienoic acid (16:3); oleic acid (18:1); linoleic acid (18:2); or linolenic acid (18:3). The SQDG were identified by typical SQDG fragments [8], which did not provide further information about the fatty acids bound. It is not possible to assign fatty acids to the sn-1 or sn-2 positions. Consequently, there are different possible fatty acid combinations for several SQDG. Furthermore, the unsaturation of the fatty acids also leads to additional possible combinations: e.g., 16:2/16:0 or 16:1/16:1 (*m/z* 789); 18:3/16:1 or 18:1/16:3 or 16:2/18:2 (*m/z* 813); and 18:3/18:3 (*m/z* 837). In contrast, the less often formed SQDG had myristic acid (14:0) and/or palmitic acid (16:0) linked. In detail: 14:0/16:0 (*m/z* 765), and 16:0/16:0 (*m/z* 793). Overall, an increase in the formation of SQDG with unsaturated fatty acids and a decrease in SQDG with saturated fatty acids were identified. The percentual distribution of the individual SQDG must also be considered. This illustrates the real distribution in the membrane and indicates which individual SQDG are the most common. The main SQDG derivatives in the lettuce samples were 793, 815, 817, 839, and 841. These five main SQDG represented around 80% of the total SQDG content in each lettuce, regardless of the S fertilization. Their proportional occurrence is shown in Figure 2.

SQDG derivative 815 was the main sulfolipid present in all of the S application levels. A percentage rise of 7% of the occurrences at S2 compared to S0 could be calculated and led to a total composition of 42% at S2. Significantly, higher quantities of SQDG derivatives 815 (18:3/16:0 or 18:0/16:3), 813, and 837 were found, while lower quantities of SQDG derivatives 765 and 793 were detected in the lettuce exposed to optimal or excessive S fertilization. However, no further significant differences were observed between the contents of the before mentioned SQDG when the S supply was increased from 1 mM to 1.5 mM (Figure 3). In total, a decrease in saturated fatty acids (3%) and a corresponding increase in unsaturated fatty acids (3%) were identified.

#### 2.1.2. Determination of Fatty Acids in the Galactolipids

The change in the galactolipid content was estimated on the basis of the fatty acids that were esterified. However, in such an approach differentiation between DGDG and MGDG is not possible. A 2.3-fold increase (from 0.3 µg g^−1^ DM to 0.7 µg g^−1^ DM) in the total fatty acids was detected as a result of the application of S2 in comparison to S0. Furthermore, when comparing S1 and S2 levels, a further trend toward a rise in concentration can be seen (Figure 4).

An identification of the following fatty acids was possible: palmitic acid, stearic acid, linoleic, and linolenic acid. A clear trend toward an increased formation of unsaturated fatty acids was identified. In total, an increase of 25% of 18:3 and a decrease of 18% of 16:0 was observed from the S0 to S2 treatments. These changes led to a proportion of 72% unsaturated and 28% saturated fatty acids linked to the chargeless galactolipids at a higher S level (S2) (Figure 5).

### 2.2. Total P and S Contents

The total P and S contents in the negatively charged lipid fraction 2 might indicate trends in the formation of the S and P containing polar lipids. By increasing the S treatment to lettuce plants, a significant increase in the S content was observed, in addition to no significant changes in total P content in lipid fraction 2. Overall, the P content was more than 20 times higher in the lettuce plants grown under S limiting conditions (S0) and seven times higher in response to S2 treatment (Figure 6). Due to the presence of phospholipids in various membranes, a higher content of P could have been expected.

### 2.3. Impact of Various S Levels on Pigment Concentration

In this experiment, the impact of various S treatments on pigment concentration was investigated, with a focus on chlorophyll (Chl) a, Chl b, and carotenoids (Car). These pigments are located in the so-called light-harvesting complexes (LHC), which are embedded in the thylakoid membrane. The effect on pigments can be an indirect indicator of an alteration in photosynthesis and is associated with plants health [26].

A significant increase (*p* < 0.001) in the concentrations of Chl a, Chl b, and Car was observed, in addition to a rise in the total pigment level in lettuce plants treated with elevated S concentrations (S2) (Table 2). The highest significant accumulations were detected in lettuce grown under S2 in contrast to S0, namely 96% (1.4 mg g^−1^ DM to 2.8 mg g^−1^ DM), 93% (0.6 mg g^−1^ DM to 1.1 mg g^−1^ DM), and 90% (0.8 mg g^−1^ DM to 1.7 mg g^−1^ DM) in Chl a, Chl b, and Car, respectively.

Overall, almost a doubling of the total pigment concentration was observed in the lettuce plants in comparison between the S0 and the S2 treatment. The Chl concentrations in lettuce grown under S deficiency medium, were in line with the results of Roosta et al. [27], Kalaji et al. [28], and Fu et al. [29], who discovered that under nutrient insufficiency (K, Mg, Ca, N, and S), different parts of the photosynthetic apparatus could be affected, including Chl. In contrast, Car was accumulated in the green multi-leaf lettuce under similar conditions.

With regard to the pigment profile, the same compositions were found in all samples, namely 5:2:3 Chl a:Chl b:Car. Furthermore, no pigment could be identified that might be preferentially formed by a change in S availability in the growing media.

## 3. Discussion

The aim of the current study was to identify whether S fertilization has an impact on the content, composition, and transformation of sulfolipids in the chloroplast membranes of green multi-leaf lettuce. A significant increase in the total SQDG content, as well as a change in the SQDG composition, were detected. The formation of some individual SQDG was affected by the S treatments, toward either a preferred or less favored formation. The percentual content of SQDG with unsaturated fatty acids was enhanced by a higher S supply; in contrast, SQDG with saturated fatty acids were formed frequently. Consequently, a higher level of total unsaturated fatty acids was determined. Furthermore, fatty acids of galactolipids were determined to show whether the trend toward a higher degree of unsaturation of sulfolipids could be confirmed for galactolipids as well. A correlation of the different S treatments and a change in content, as well as an alteration of the composition of individual lipids, was observed. Additionally, analysis of the pigments indicated potential impacts on photosynthesis.

The total SQDG content was enhanced up to six-fold in the lettuce plants under varied treatments (at S2 to S0), whereas the total fatty acids of chargeless lipids were affected by the more than doubling of the content. A ratio of MGDG and DGDG could not be determined by this method. The total formation of SQDG seemed to be more likely in comparison to the formation of galactolipids. Sulfolipids and PG are anionic thylakoid membrane lipids. Consequently, the increase in SQDG content could result in a change in chloroplast membrane lipid composition, which might lead to a higher percentage of anionic lipids. PG can be characterized as a glycerol-based phospholipid with a negative charge and can be found exclusively in chloroplast membranes. A higher proportion of anionic thylakoid membrane lipids may induce a change in the charge in the membrane. In contrast to PG, phospholipids are located in various plant membranes. In this context, the analyzed total P content of the charged lipid fraction does not only include PG, but also other phospholipids. Thus, the exact content of PG cannot be determined, but only the total phospholipid content in the sum of all membranes. With regard to the findings of the present study, the total P content showed no significant differences in response to the different S levels. The occurrence of phospholipids in several membranes implies that the P content is strongly influenced by non-photosynthetic membranes. Furthermore, the total P content does not provide any information about the charge of the thylakoid membrane. However, total P content could be an indicator of the unchanged concentrations of phospholipids in all membranes, even when the S content is changed. In contrast, the increase in the total S content is correlated with increased total SQDG concentrations. Plants seem to use SQDG as storage for S [18]. Through sufficient or excessive availability of S, lettuce could probably store S in the form of SQDG.

By using various mutants of plants, an attempt was made to understand the function of PG and SQDG in the context of photosynthesis. Under P deprivation conditions, SQDG were able to compensate PG levels to a certain extent [30]. Whereas PG seemed to be indispensable for photosynthesis, SQDG can be replaced by PG with only subtle growth impairments [31]. However, the total increase in the chloroplast membrane lipids and a higher quantity of anionic lipids in the form of SQDG probably affected the structure and function of the thylakoid membrane.

The impact of SQDG on the structure and function of the thylakoid membranes and photosynthesis is still not clearly understood, but they seem to be necessary to maintain optimal PS II activity. In particular, the conformation of the PS II core complex is altered in mutants of plants, with a lack of SQDG [32,33,34,35,36]. Pigments are rearranged in LHC. They capture the light and transfer the formed excitation energy to the reaction centers. Under the condition of high light intensity, light dissipation plays an important role in avoiding damage to the photosynthetic apparatus. The entire mechanism is still unknown. Conformation change to the LHC of PS II, and an associated change in the positions of pigments to one another with a resulting change in the intra-molecular organization, seem to play a protective role [37]. Schaller et al. [38] studied the aggregation of LHC II in vitro in dependency of thylakoid membrane lipids. Lipid-depleted LHC II complexes showed a higher aggregation. Through the re-addition of the thylakoid lipids, a change in the spectroscopic properties and indications of a structural change were described. In particular, anionic lipids led to a disaggregation of the LHC II and an intensified increase in the Chl fluorescence [38]. This seems to be important for regulating UV protection [38]. In contrast, chargeless galactolipids led to rearrangements within LHC II aggregates. Additionally, lipid concentration influenced Chl a fluorescence. The arrangement of membrane proteins was also affected by the concentration of lipids, but they did not interfere with protein function [38]. It is not possible to compare in vitro experiments in general with the behavior of plants in vivo, but in plants, similar reactions may occur, and it is still suggested that the lipids have an impact on the function of LHC II and the macro-organization in vivo [39]. The stronger increase in the sulfolipid content could result in a change in aggregation of the LHC and an alteration in Chl fluorescence. SQDG are also important for stabilizing the arrangement of proteins involved in photosynthesis [40]. A change in the composition of the membrane lipids could have an impact on the stabilization as well.

In the present study, a stronger increase in the total sulfolipids than galactolipids was observed. There was a six-fold increase in the sulfolipid content, instead of doubling as in the case of galactolipids, and the resulting increase in the total anionic lipids could lead to an intensified disarrangement of the LHC. This could then result in a change in the thylakoid membrane organization, an effect of photosynthesis function, and a modification of the UV protection.

In addition to the enhancement of lipid content by increased S treatments, a clear change in the unsaturation of fatty acids is detectable. The degree of unsaturation affects the structure and function of a membrane. SQDG showed a preferred formation of several SQDG with linked polyunsaturated fatty acids and a decrease in two SQDG with linked saturated fatty acids. Why these few SQDG were affected more than the others cannot yet be answered comprehensively. However, it was shown for the first time that not all SQDG are affected equally, as only a few individual SQDG were affected more than others. So far, there were no data about an alteration of individual sulfolipids. The observed trends toward a higher quantity of unsaturated fatty acids in the glycolipids were detected better in the galactolipids. A clear formation of linolenic acid linked to MGDG or DGDG was shown by an increased S supply. The enhancement of the total percentage of linolenic acid and the decrease in palmitic acid could indicate an affected membrane structure and function. Belkhadi et al. [41] suggested that high amounts of polyunsaturated fatty acids in chloroplasts are important for the geometry of lipid molecules and/or to maintain the degree of fluidity. The alteration of the saturation of fatty acids relates to biotic or abiotic stress. Under abiotic stress conditions, the maintenance of the protein function involved in photosynthesis is crucial. There are different abiotic stress adaption strategies that use alterations of fatty acids. While an enhanced formation of unsaturated fatty acids is favored during chilling stress [42,43], mainly saturated fatty acids are formed during drought [44]. In addition to temperature, other abiotic stress factors, such as salt and heavy metals, have an impact on membrane lipids. A decline in linolenic acids can be seen due to salt stress [45]. Zhang et al. [46] showed that an overexpression of ω-3 desaturases higher levels of linolenic acids in transgenic tobacco cells. As a result, this overexpression led to a better adaption to salt and drought. Unsaturated fatty acids might reduce the photoinhibition of PS II and PS I, and therefore ensure better adaption to stress [47]. Consequently, the decline in linolenic acids in plants could indicate damage caused by abiotic stress. The ability to maintain or to adapt fatty acid unsaturation and/or the present concentration of unsaturated fatty acids influences the capacity of plants towards stress [48,49]. In the case of heavy metal stress, a reduction in unsaturated fatty acids was detected as well [50]. The exposure to heavy metals leads to an enhanced formation of reactive oxygen species (ROS) and lipid peroxidation. Damage and loss of the integrity of the membrane can be the consequence [51].

A change in the degree of unsaturation seems to be more altered in galactolipids than in sulfolipids. At around 75%, galactolipids represent the largest part of the thylakoid lipids. Alterations of the galactolipids may be more relevant to the function and structure of the thylakoid membrane. With an enhancement of around 25% of linolenic acid, modified parameters of the membrane can be expected.

In green multi-leaf lettuce, lower ratios of unsaturated fatty acids can be an indicator for abiotic stress caused under S deprivation (S0). Lipid peroxidation is associated with abiotic stress and could be one reason for the decline in the unsaturated fatty acids; this can explain the reduced linolenic acid levels in lettuce grown under S-limiting conditions (S0), in contrast to the S1 and S2 treatments, respectively. A reason for the increase in the linolenic acid in lettuce grown under S1 and S2 treatments could be a continued modification of the lipid peroxidation. The further increase in S fertilization (at S1 and S2) could cause a further change and alter lipid oxidation; thus, its effects on fatty acids in green multi-leaf lettuce could be due to an increased nutrient supply. Zhou et al. [52] showed that malondialdehyde (the most prominent marker of lipid peroxidation) was lower in S-fertilized wheat under drought stress. In the scavenging of ROS (for diminishing lipid peroxidation) various mechanisms are involved. In addition to superoxide dismutase, tocopherol, GSH, plant polyphenols, and other antioxidants might have an impact on the suppression of lipid peroxidation. Fatma et al. [53] showed the impact of S fertilization on GSH content. The addition of S led to an increase in GSH in non-stressed mustard plants. Through additional salt stress exposure, a further increase in GSH concentration was demonstrated. An excess in S is more effective in reducing oxidative stress and is able to minimize the negative effects on photosynthesis by salt exposure [53]. The positive effect of S fertilization in the context of ROS could influence lipid peroxidation in stressless conditions as well. Higher levels of linolenic acids could be the result. An enhancement of linolenic acids could help lettuce to better adapt to further stress conditions.

In the present study, S fertilization also had a positive effect on pigment content. It was confirmed that S influences photosynthesis [52]. In response to S deficiency conditions, a decrease in Chl content, photosynthetic CO_2_ uptake, photoreduction of ferricyanide, and CO_2_ assimilation by ribulose diphosphate carboxylase were previously indicated [54]. Additionally, the findings showed that an increase in S supply from S1 to S2 also resulted in a further accumulation of the total pigment content in lettuce. The photosynthetic pigments are associated with the photosynthesis process, for which the observed increase in concentration could have an indirect impact. The Chl a/b ratio illustrates the impacts on PS I or II. In the present study, the Chl a/b ratio did not change, suggesting that the PSI:PSII ratio did not change as well. These findings are in accordance with the results discovered by Lunde et al. [11] who showed that S deprivation leads to a reduction in Chl a and b, but no change in the ratio and, consequently, no change in the PS ratio. In comparison to S-limiting conditions, higher concentrations of pigments were identified. These results are in line with Zhou et al. [52], who indicated that elevated levels of photosynthetic pigments were higher in S fertilized wheat.

Whether the increase in the content of pigments, galactolipids, and sulfolipids affects the composition of LHC or the thylakoid membrane structure, respectively, cannot be concluded so far. The alteration of membrane lipids using S fertilization seems to be a prominent possibility for stress adaptation to different environmental conditions.

## 4. Materials and Methods

### 4.1. Chemicals and Materials

The SQDG derivative 816 (2-O-hexadecanoyl-1-O-(9Z,12Z,15Z-octadecatrienoyl) glycerol-3-(6-deoxy-6-sulfo-α-D-glucopyranoside)) was purchased form Avanti Polar Lipids Inc. (Alabaster, AL, USA). Acetonitrile, chloroform, methanol, and trimethylsulfonium hydroxide (TMSH) were purchased from Carl Roth GmbH & Co KG (Karlsruhe, Germany). Ammonium acetate, copper sulfate, Supelco™ FAME MIX C_8_-C_24_, and potassium chloride were purchased from Sigma-Aldrich GmbH (Munich, Germany). Ammonia (25%), and nitric acid (65%, analytical-reagent-grade), phosphorus (P), S (ICP standard), and tert-butylmethylether (MTBE) were purchased from Merck KGaA (Darmstadt, Germany). All aqueous solutions were prepared with deionized water, generated by a Purelab flex water purification system (Veolia Water Technologies Deutschland GmbH, Celle, Germany). Amino-phase (NH_2_) solid phase extraction cartridges (6 mL, 500 mg) were purchased from Macherey-Nagel GmbH & Co. KG (Düren, Germany).

### 4.2. Plant Material and Growth Conditions

The experiment was carried out in January 2021 and was conducted in a greenhouse (Institute of Plant Nutrition and Soil Science, University of Kiel, Kiel, Germany). The green multi-leaf lettuce cultivar ‘Hawking RZ’ was hydroponically cultivated. To ensure a better germination rate, seeds were placed in sandwich blots in a 1:5 diluted 10 mM CaSO_4_ solution. The seeds were germinated in a climatic growth chamber (with 14 h of light, 23 °C during the day, 14 °C at night, and 39% humidity) for 14 days. The seedlings were planted individually in 10 L-black containers that were randomly arranged and kept under standard greenhouse conditions (18 °C day/14 °C night cycle and a 14 h photoperiod). The basal nutrient solution was made up of 2 mM Ca(NO_3_)_2_, 0.5 mM NH_4_H_2_PO_4_, 0.5 mM MgCl_2_, 2 mM KNO_3_, and a micronutrient solution of 60 μM Fe-EDTA, 10 μM H_3_BO_3_, 2 μM MnSO_4_, 0.5 μM ZnSO_4_, 0.3 μM CuSO_4_, and 0.01 μM (NH_4_)_2_Mo_7_O_24_. The pH was adjusted from 6.0 to 6.5. The nutrient solution was changed weekly. The treatments were laid out in a completely randomized design. Three S levels (S0, S1, and S2) were applied in the nutrient solution. The tested S treatments were as follows: (1) control (S0: 0 mM K_2_SO_4_); (2) S1: 1 mM K_2_SO_4_; and (3) S2: 1.5 mM K_2_SO_4_. The lettuce plants were harvested on the 55th day of the experiment. The heads of the lettuce were separated from the roots. The lettuce heads were washed with distilled water, frozen in liquid nitrogen, and subsequently dried at −53 °C in a freeze dryer (Gamma 1-20, Martin Christ Gefriertrocknungsanlagen GmbH, Osterode am Harz, Germany). The dried lettuce heads were ground to a fine powder and kept for further analyses.

### 4.3. Lipid Extraction Procedure

Galactolipids and sulfolipids were extracted with a chloroform/methanol (3/2, *v*/*v*) mix according to the method described by Fischer et al. [8] Extraction from the lettuce samples was improved by a ball mill (frequency 25 Hz, 7 balls ø = 1.5 cm; Retsch MM 400, Retsch GmbH, Haan, Germany).

### 4.4. Clean-Up and Separation of the Lipids by Solid Phase Extraction

Separation of the chargeless galactolipids and negatively charged sulfolipids were achieved by a simple solid phase extraction with anion exchange function, following a protocol described by Fischer et al. [8], with slight modifications. NH_2_-SPE cartridges were used for separating the different charged lipids. The sample extract was loaded onto the cartridge. Galactolipids were evaluated with 10 mL chloroform/methanol (9/1, *v*/*v*) and 10 mL chloroform/methanol (5/5, *v*/*v*), and collected in a 50 mL-tube (‘fraction 1’). SQDG were eluted with 10 mL chloroform/methanol (4/1, *v*/*v*), containing 100 mM ammonium acetate and 2% NH_3_ (‘fraction 2’). Fraction 2 was mixed with 2 mL 0.9% potassium chloride solution. After vortexing, ultra-sonification for 10 min, and centrifugation (10 min, 12,000× *g*), the organic phase was separated. Solutions were dried with a gaseous stream of N_2_, and fraction 1 was resolved in 3 mL methanol (‘solution 1’ for the determination of MGDG, and DGDG), and fraction 2 was resolved in 1 mL MeOH (‘solution 2’ for the determination of SQDG).

### 4.5. Determination of Sulfolipids Using LC-ESI-MS/MS

Mass spectrometer parameters were according to Fischer et al. [8], with slight modifications. LC-MS analytic was performed on Agilent 1260 Infinity II HPLC-system (Agilent Technologies Deutschland GmbH, Waldbronn, Germany) coupled with 5500 QTrap triple quadrupole MS/MS system (AB Sciex Germany GmbH, Darmstadt, Germany). Separation of ‘solution 2’ (SQDG) was achieved with Kinetex^®^ C18 column (5 μm, 100 Å, 150 mm × 2.1 mm; Phenomenex Ltd., Aschaffenburg, Germany), using a constant flow rate of 0.3 mL/min. Eluent A was water and eluent B was acetonitrile/water (9/1; *v*/*v*), each with 10 mM ammonium acetate. The elution started with 3% eluent B for 2 min and linearly increased to 99% eluent B within 4 min, which was kept constant for 32 min. Then, the composition was readjusted to 3% eluent B within 2 min, followed by 4 min of re-equilibration. The SQDG derivative 816 was used as standard in the concentration range of 0.1–1 µg mL^−1^.

### 4.6. GC-MSD Determination of Galactolipids on the Basis of their Fatty Acids Profile

The fatty acids of galactolipids were analyzed as fatty acid methyl esters (FAME). A total of 20 µL of the extract ‘solution 1’ was dried under N_2_ and resolved in 40 µL MTBE. A total of 40 µL TMSH was used as methylation reagent. Derivatization was performed at room temperature for 1 h. Nonadecanoic acid methyl ester was used as internal standard. The GC-MS analytic was performed on an Agilent Technologies 7890 gas chromatograph coupled to Agilent 5975 mass selective detector (MSD) (Agilent Technologies Inc. Santa Clara, CA, USA). For separation of FAME a HP-5 column (30 m × 0.25 µm × 0.32 mm, Agilent Technologies Inc., Santa Clara, CA, USA) was used as stationary phase and helium as carrier gas. The temperature was programmed from 100 °C to 240 °C at 8 °C/min ramp rate. The injector and GC/MS interface temperatures were maintained at 290 °C and 300 °C, respectively. FAME MIX C_8_–C_24_ were used for identification and quantification.

### 4.7. Determination of the Total P and S Content in the SQDG Extracts

The SQDG extracts (‘solution 2’) were transferred in a Kjeldahl flask and dried with a gaseous stream of N_2_. They were then digested in 5 mL 65% HNO_3_ at 200 °C for 30 min, cooled down, and filled with bidest water to 25 mL. Determination of P and S was performed using ICP-OES (Spectro Arcos, Spectro Analytical Instruments GmbH, Kleve, Germany) with EOP torch type. Operating parameters were: RF generator (1400 W); Plasma gas flow, auxiliary gas flow, and nebulizer gas flow were 12.0, 1.0, and 1.0 L/min, respectively. ICP-standards (P and S) were used in concentration range of 0.1–10 mg L^−1^, respectively.

### 4.8. Determination of Total Chlorophyll a, b and Carotenoids

Determination of chlorophyll a, b, and carotenoid concentration was performed according to Ghosh et al. [26], with slight modifications. A total of 100 mg of freeze-dried material were extracted by 10 mL acetone/water (4/1, *v*/*v*) in a ball mill (frequency 25 Hz, 3 balls ø = 1.5 cm) for 5 min, following 15 min of ultra-sonification and centrifugation (10 min, 12,000× *g*, RT), the supernatant was removed and collected in a 50 mL measuring cylinder. Extraction was repeated three times and then the solution was filled up to 50 mL. Absorption was measured at 470 nm, 645 nm, and 663 nm against the solvent (80% acetone) at a UV-vis spectrometer (Thermo Fisher Scientific Inc., Waltham, MA, USA).

### 4.9. Statistical Analysis

Statistical analysis was performed with OriginPro 2021 (OriginLab Corp., Northampton, MA, USA). Differences among the application levels were evaluated using two-way ANOVA test and the post-hoc Scheffe test was selected to evaluate probable differences among groups. A level of *p* < 0.05 was considered to be significant. The results were presented as means and the standard errors of means.

## 5. Conclusions

Many studies have already shown a change in chloroplast membrane lipids due to abiotic stress, e.g., a lack of macronutrients. The focus was mainly on the total content, as well as the change in the degree of unsaturation of the fatty acids linked to membrane lipids. In this study, an influence on individual SQDG and the fatty acids of the galactolipids could be shown as a result of sulfur nutrition. Not only a deficiency but also an excess of S has a strong influence on chloroplast membrane lipids and thus hypothetically on the function and structure of the chloroplast membranes. The trend of a change of the degree of unsaturation towards a higher level of unsaturated fatty acids linked to SQDG and/or galactolipids could be a marker for a modified health status of green multi-leaf lettuce, and might consequently also have a positive effect against other stress factors. There is also an influence on the pigment content, which in total might have an indirect influence on photosynthesis, which needs to be checked in detail in future studies. At this point it is expected that the results might result in a better photosynthesis capability of green multi-leaf lettuce and enable chances for optimum plant nutrition, in the case of lettuce.

## Figures and Tables

**Figure 1 plants-11-01342-f001:**
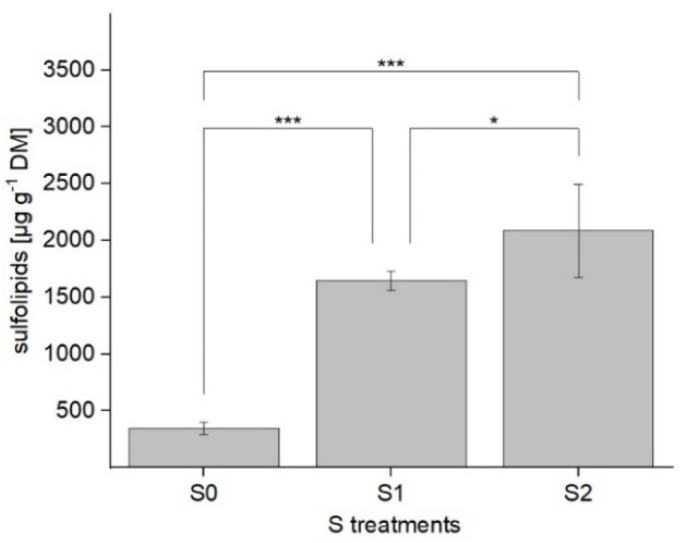
Total sulfolipid content (µg g^−1^ DM) in green multi-leaf lettuce grown under different S treatments. Values represent the mean ± SD of independent replicates of lettuce treated with S0: 0 mM (*n =* 3), S1: 1 mM (*n =* 6), and S2: 1.5 mM (*n* = 4) K_2_SO_4_, respectively. Asterisks indicate the different levels of significance (* *p* < 0.05, *** *p* < 0.001).

**Figure 2 plants-11-01342-f002:**
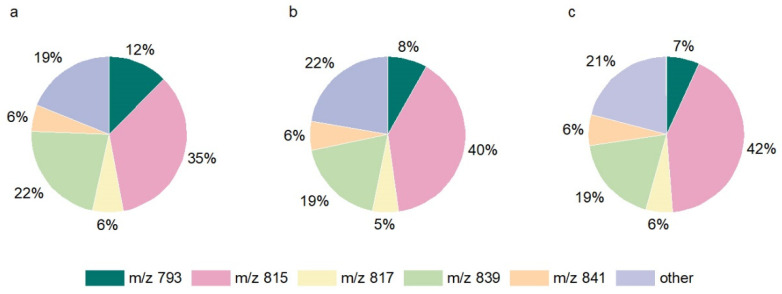
Distribution of the most abundant sulfoquinovosyl diacylglycerol derivatives (SQDG) (%) in relation to total SQDG content in lettuce. Plants were treated with (**a**) S0: 0 mM; (**b**) S1: 1 mM; and (**c**) S2: 1.5 mM K_2_SO_4_. Individual SQDG derivatives were displayed with their mass to charge ratio (*m/z*).

**Figure 3 plants-11-01342-f003:**
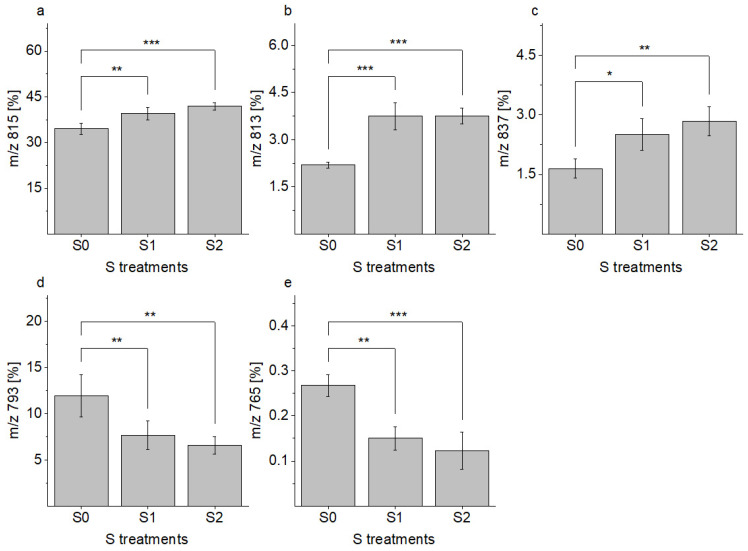
Content (%) of individual sulfoquinovosyl diacylglycerol derivatives, with *m/z* of (**a**) 815; (**b**) 813; (**c**) 837; (**d**) 793; and (**e**) 765, in green multi-leaf lettuce grown under different S treatments. Values represent the mean ± SD of of independent replicates of lettuce treated with S0: 0 mM (*n =* 3), S1: 1 mM (*n =* 6), and S2: 1.5 mM (*n =* 4) K_2_SO_4_, respectively. Asterisks indicate the different levels of significance (* *p* < 0.05, ** *p* < 0.01, *** *p* < 0.001).

**Figure 4 plants-11-01342-f004:**
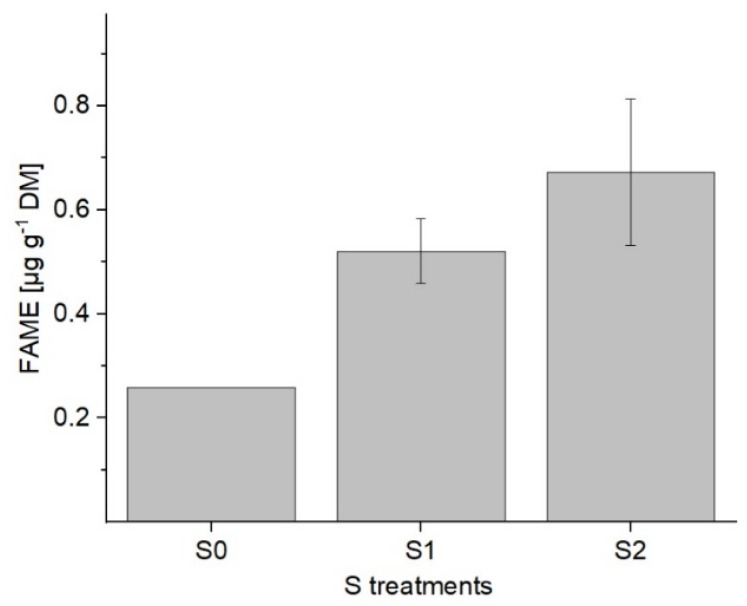
Total fatty acid content (µg g^−1^ DM), determined as methyl esters (FAME) in green multi-leaf lettuce grown under different S treatments. Values represent the mean ± SD of independent replicates of lettuce treated with S0: 0 mM (*n* = 1), S1: 1 mM (*n* = 2), and S2: 1.5 mM (*n* = 2) K_2_SO_4_, respectively.

**Figure 5 plants-11-01342-f005:**
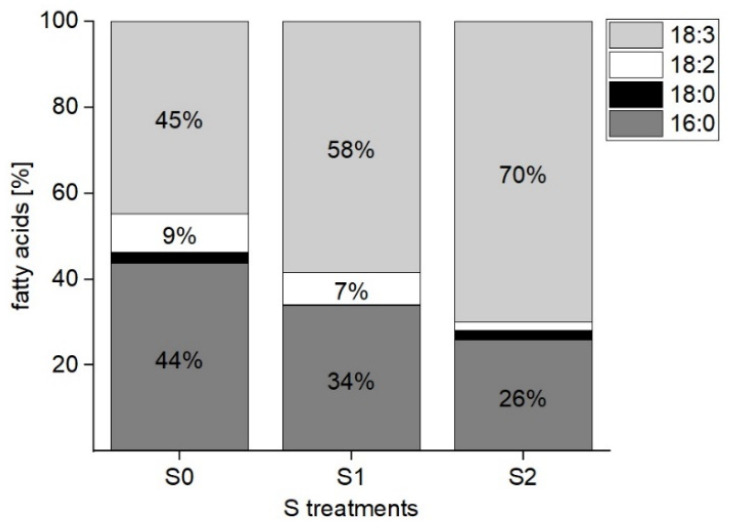
Composition (%) of fatty acids in green multi-leaf lettuce grown under different S treatments (S0: 0 mM, S1: 1 mM, and S2: 1.5 mM K_2_SO_4_, respectively).

**Figure 6 plants-11-01342-f006:**
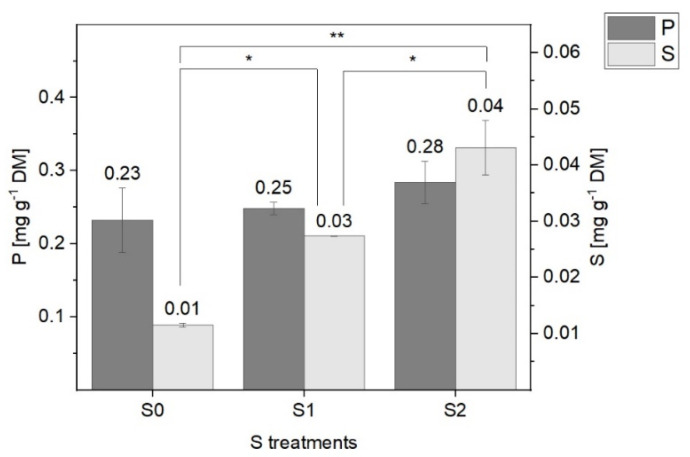
Total P and S contents (mg g^−1^ DM) in the negatively charged lipid fraction of green multi-leaf lettuce grown under different S. Values represent the mean ± SD of two independent replicates of lettuce treated with S0: 0 mM (*n =* 2), S1: 1 mM (*n =* 2), and S2: 1.5 mM (*n =* 2) K_2_SO_4_, respectively. Asterics indicate the significant differences (* *p* < 0.05, ** *p* < 0.01).

**Table 1 plants-11-01342-t001:** Individual SQDG content (%) in relation to total SQDG content in lettuce plants grown under different S treatments. Values represent the content of individual SQDG (%) in green multi-leaf lettuce treated with S0: 0 mM, S1: 1 mM, and S2: 1.5 mM K_2_SO_4_, respectively. Different letters suggested significant differences (*p* < 0.05). Abbreviations: SQDG: sulfoquinovosyl diacylglycerol derivatives, *m/z*: mass to charge ratio.

Individual SQDG *m/z*	SQDG Content at Varied S Treatments (mM)			Enrichment Factor (c_1.5mM_/c_0mM_)
	S0 (%) ^1^	S1 (%) ^2^	S2 (%) ^3^	
765	0.3 *^a,b^*	0.2 *^a^*	0.1 *^b^*	3.1
787	0.2	0.3	0.3	7.9
789	0.2	0.3	0.3	9.2
791	0.6	0.6	0.5	5.0
792	1.1	1.2	1.1	6.0
793	12.4 *^a^*	8.3 *^b^*	6.9 *^b^*	3.4
813	2.2 *^a^*	3.7 *^b^*	3.8 *^b^*	10.4
815	34.5 *^a^*	39.5 *^b^*	41.9 *^b^*	7.3
817	6.4	5.4	5.5	5.2
819	1.5	1.9	1.8	7.3
821	0.2	0.3	0.3	7.8
837	1.6 *^a,b^*	2.5 *^a^*	2.8 *^b^*	10.4
839	22.2	18.5	18.5	5.0
841	5.6	6.0	6.4	6.9
843	2.6	2.7	2.4	5.4
845	1.4	1.2	1.1	4.8
847	0.9	0.9	0.6	4.1
849	0.3	0.3	0.2	2.9
855	2.1	2.4	2.0	5.9
555	3.5	3.6	3.5	5.9

^1^ Results of three different biological replicates; ^2^ Results of six different biological replicates; ^3^ Results of four different biological replicates.

**Table 2 plants-11-01342-t002:** Total chlorophyll (Chl) a, Chl b, and carotenoid (Car) contents (mg g^−1^ DM) in leaves of green multi-leaf lettuce grown under different S treatments. Values represent the mean ± SD in green multi-leaf lettuce treated with S0: 0 mM, S1: 1 mM, and S2: 1.5 mM K_2_SO_4_, respectively. Different letters suggested significant differences (*p* < 0.001). Abbreviations: Chl: chlorophyll; Car: carotenoids.

S Treatment Level (mM)	Chl a (mg g^−1^ DM) ^1^	Chl b (mg g^−1^ DM) ^2^	Chl a/b Ratio	Car (mg g^−1^ DM) ^3^
S0	1.355 ± 0.094 *^a^*	0.559 ± 0.051 *^b^*	2.424	0.842 ± 0.038 *^c^*
S1	2.277 ± 0.083 *^a^*	1.019 ± 0.049 *^b^*	2.235	1.298 ± 0.013 *^b^*
S2	2.781 ± 0.28 *^a^*	1.131 ± 0.108 *^b^*	2.459	1.659 ± 0.047 *^c^*

^1^ Results of three different biological replicates analyzed in technical triplicates; ^2^ Results of six different biological replicates analyzed in technical triplicates; ^3^ Results of four different biological replicates analyzed in technical triplicates.

## Data Availability

The data sets presented in this study are available upon request from the corresponding author.

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
