# Peer review of "Alterations of Content and Composition of Individual Sulfolipids, and Change of Fatty Acids Profile of Galactolipids in Lettuce Plants (Lactuca sativa L.) Grown under Sulfur Nutrition"

_plants, 2022, doi:10.3390/plants11101342_

Round 1

Reviewer 1 Report

The work presented for review is an interesting approach to the problem of sulfur feeding of plants. The paper presents many valuable chemical analyzes of sulfur, hydroponics, sulfolipids, fatty acid, chlorophyll a, b and carotenoids determinations. The work is written in the correct language and graphically legible. However, doubts are raised by: 1. The authors give abiotic stress as key words, as you can guess it is salt stress caused by a high dose of sulfur, but there are no determinations of salt concentration in the medium and its pH. 2. A single factor experiment where only 2 doses of sulfur were used and no sulfur controls were used. So the whole experience is based on only two doses of sulfur, this is a very narrow scope of research. 3. There is no information on how many years of experience these have been. Only annuals Please indicate the years of the study. 4. The work is based on 53 literature items. Unfortunately, 34 items are older than 10 years and as many as 15 are older than 20 years. 5. Conclusions are unfortunately not well written and insufficient. There is no specific information about the effect of sulfur doses on the parameters tested in plants, e.g. it was not determined whether the effect was positive or negative on the photosynthesis process (line 521). The work is ready for publication, however, after completing the information and redrafting conclusions and supplementing the literature with the latest items.

Reviewer 2 Report

The manuscript entitled "Alterations of Content and Composition of Individual Sulfolipids, and Change of Fatty Acids Profile of Galactolipids in Lettuce Plants (Lactuca sativa L.) Grown under Sulfur Nutrition" written by Korber et al., was very well written an original. It emphasizes the role of S containing compounds including sulfolipids in modulating physiological adjustment mechanisms to improve tolerance ability to various abiotic stresses in plants and consequently, plant food quality. I suggest the acceptace of the paper in the present form.

Reviewer 3 Report

This is a well-written manuscript that reported the effects of sulfur nutrition on the content and composition of individual sulfoquinovosyldiacylglycerol (SQDG) derivatives in green multi-leaf lettuce plants. The study provides insights into the crucial role of sulfur on plant pigments and photosynthetic apparatus. I highly recommend this manuscript for publication.

Round 2

Reviewer 1 Report

Accept in present form

Author Response

No further comments.